# *Drosophila melanogaster* as a Model to Study Fragile X-Associated Disorders

**DOI:** 10.3390/genes14010087

**Published:** 2022-12-28

**Authors:** Jelena Trajković, Vedrana Makevic, Milica Pesic, Sofija Pavković-Lučić, Sara Milojevic, Smiljana Cvjetkovic, Randi Hagerman, Dejan B. Budimirovic, Dragana Protic

**Affiliations:** 1Faculty of Biology, University of Belgrade, 11000 Belgrade, Serbia; 2Department of Pathophysiology, Faculty of Medicine, University of Belgrade, 11000 Belgrade, Serbia; 3Institute of Human Genetics, Faculty of Medicine, University of Belgrade, 11000 Belgrade, Serbia; 4Department of Pharmacology, Clinical Pharmacology and Toxicology, Faculty of Medicine, University of Belgrade, 11000 Belgrade, Serbia; 5Department of Humanities, Faculty of Medicine, University of Belgrade, 11000 Belgrade, Serbia; 6Medical Investigation of Neurodevelopmental Disorders (MIND) Institute, University of California Davis, 2825 50th Street, Sacramento, CA 95817, USA; 7Department of Pediatrics, University of California Davis School of Medicine, Sacramento, CA 95817, USA; 8Department of Psychiatry, Fragile X Clinic, Kennedy Krieger Institute, Baltimore, MD 21205, USA; 9Department of Psychiatry & Behavioral Sciences-Child Psychiatry, Johns Hopkins School of Medicine, Baltimore, MD 21205, USA

**Keywords:** Fragile X syndrome, FXTAS, *FMR1* gene, FMRP, *Drosophila melanogaster*

## Abstract

Fragile X syndrome (FXS) is a global neurodevelopmental disorder caused by the expansion of CGG trinucleotide repeats (≥200) in the Fragile X Messenger Ribonucleoprotein 1 (*FMR1*) gene. FXS is the hallmark of Fragile X-associated disorders (FXD) and the most common monogenic cause of inherited intellectual disability and autism spectrum disorder. There are several animal models used to study FXS. In the FXS model of *Drosophila*, the only ortholog of *FMR1*, *dfmr1*, is mutated so that its protein is missing. This model has several relevant phenotypes, including defects in the circadian output pathway, sleep problems, memory deficits in the conditioned courtship and olfactory conditioning paradigms, deficits in social interaction, and deficits in neuronal development. In addition to FXS, a model of another FXD, Fragile X-associated tremor/ataxia syndrome (FXTAS), has also been established in *Drosophila.* This review summarizes many years of research on FXD in *Drosophila* models.

## 1. Introduction

When Morgan (1910) began his pioneering experiments in genetics in the early 1900s while working with fruit flies, he probably had no idea how important they would become over time, including studying the molecular pathogenesis of human diseases. As a model system, many advantages of the fruit fly, *D. melanogaster,* have made it the most important eukaryotic organism for understanding basic genetic principles, inheritance mechanisms, chromosomes and genes, and mutagenesis. These advantages include a small body size (2–3 mm), easy and inexpensive cultivation and maintenance in the laboratory, a large number of offspring per mating (~100 eggs per day), and a rapid life cycle (about ten days at 25 °C) [1]. In addition, the fruit fly, *D. melanogaster*, is an excellent example of the 3R principle (Replacement, Refinement, Reduction), which replaces the use of higher laboratory animals in research studies [2]. The 3R principle is based on the belief that animal species have a certain degree of intrinsic value that must be considered in order to adequately consider animal welfare [3].

The genome of *D. melanogaster* was sequenced and published in 2000 [4,5]. It has a size of about 180 Mb, of which about 120 Mb is euchromatin [4,6]. The predicted number of protein-coding genes is more than 14,000, together with about 3000 non-coding genes [7,8]. About 60% of all human genes and 75% of genes related to human diseases have their homologues in the fruit fly [9,10]. Although fruit flies and humans possess completely different anatomical features, they share similar and important cellular and molecular processes and biological pathways [11,12]. These highly conserved pathways are membrane excitability, synaptic plasticity, and neuronal signaling. In addition, classes of neurotransmitters, such as adrenergic, dopaminergic, serotonergic, and histaminergic, are highly conserved, too [13,14]. Some structures in the *Drosophila* brain also have their counterparts in mammals, such as the mushroom bodies involved in learning and memory, which correspond to the mammalian hippocampus [15]. Furthermore, because the fruit fly brain is compact and consists of more than 100,000 neurons involved in various behaviors, it is used as a model for “screening therapeutic drugs for various human neuropathies” [16].

Evolutionary conservation of gene functions has shown that some mechanisms in fruit flies apply to more complex versions of human behavioral processes [17]. In addition, the conservation of these genetic processes has also provided important insights into the underlying mechanisms of human diseases due to the alteration of typical neuronal functions [18,19]. Recent studies are also directed toward mechanisms involved in rare diseases [20]. Finally, *Drosophila* is an indispensable model organism for the study of the Fragile X Messenger Ribonucleoprotein 1 (*FMR1*) gene triplet repeats expansion in humans, clinically known as Fragile X syndrome (FXS) [21]. The *Drosophila* model has proven useful for early-stage pharmacological screening of drug candidates identified in the fly that need to be tested in a mammalian model of FXS [22].

## 2. Fragile X Syndrome

The *FMR1* gene, which encodes the *FMR1* protein (FMRP), is located on chromosome Xq27.3, and there are between 5 and 44 CGG trinucleotide repeats in the normal promoter region of the *FMR1* gene [23,24,25]. The expansion of the CGG triplet repeats (≥200) in the 5′ untranslated region (UTR) of the *FMR1* gene is a full mutation (FM). It causes FXS, the most common form of inherited intellectual disability (ID), and is a monogenic cause of autism spectrum disorder (ASD) [26]. Hypermethylation triggered by the FM of the *FMR1* gene results in transcriptional silencing and a reduction or absence of FMRP [27,28]. FXS affects approximately 1 in 5000 males and 1 in 8000 females [29].

The product of the *FMR1* gene, FMRP, is produced mainly in neurons (particularly in the cortex, Purkinje cells, and hippocampus) and testes and plays an important role in normal brain development [30]. FMRP is an RNA-binding protein that is mainly localized in the cytoplasm and carries a Nuclear Localization Signal (NLS) and a Nuclear Exportation Signal (NES) [31]. Fragile X Related 1 (FXR1) and Fragile X Related 2 (FXR2) are two paralogues that are highly homologous to FMRP [32]. FMRP has a specific mechanism of action: it enters the nucleus, interacts with pre-messenger ribonucleoprotein complexes (pre-mRNP), and brings them into the cytoplasm. The FMRP-mRNP complexes are associated with polyribosomes and play an important role in the translation of proteins in the neuronal soma and dendritic spines [33]. This protein also plays a role in post-transcriptional regulation and is a component of stress granules and P-bodies (reviewed in [34]). The FMRP-mRNPs complexes are also a component of RNA granules, where they mediate binding between mRNAs and kinesins and play a role in cell transport [35]. In the last decade, other FMRP targets have also been identified, and studies have shown that FMRP also plays a role in the microRNA (miRNA) and Piwi-interacting RNA (piRNA) pathways [21,36,37]. Some of these targets are involved in neurodevelopmental disorders such as idiopathic ASD and other neuronal pathologies [21].

As mentioned above, FMRP deficiency is associated with a wide range of neurobehavioral clinical features of FXS, which include physical, cognitive, and behavioral abnormalities (i.e., 50–60% are diagnosed with ASD) [38]. Characteristic physical features such as an elongated face, large or protruding ears, a high-arched palate, joint hypermobility, and macroorchidism at and after puberty are present in most individuals with FXS [39]. Behavioral features include shyness, social anxiety, attention deficits, hyperactivity, disturbed eye gaze, sensory overexcitation, aggressive behavior, sleep problems, hand flapping, repetitive behaviors, and obsessive-compulsive disorder [26,40,41,42,43]. Seizures occur in about 15–20% of people with FXS [44], while obesity and gastrointestinal dysfunction are diagnosed in over 30% of patients [45]. Men with FXS have an average Full Scale IQ (FSIQ) between 40 and 50 and are more severely affected than women with FXS, who have an average FSIQ between 70 and 80, although 1/3 have an IQ below 70 [46].

## 3. *Drosophila* Model for the Study of Fragile X Syndrome

Numerous animal models of FXS include *Fmr1* knock-out (KO) mice (discovered more than 20 years ago), rats, the *D. melanogaster* model of FXS, and *Fmr1* KO zebrafish [21,47]. All these models are based on the disruption or KO of the *FMR1* gene homologue. However, there are no naturally occurring animal models for FXS [22]. This review focuses on using the *Drosophila* FXS model to study FXS.

The *FMR1* homolog in *Drosophila* was identified in 2000 and designated *dfmr1* [48]. In the meantime, there have been a few different names for this homolog (for more details, please visit flybase.org, accessed on 1. December 2022), and the current official name is *Fmr1*. Nevertheless, in this review, we mark it as *dfmr1* to avoid confusion and to distinguish it from the human *FMR1* gene. *dfmr1* is 8.7 kb long, and all of its functional domains are highly conserved, with two KH domains being 35% identical and 60% similar between *dfmr1* and human *FMR1* [48]. The encoded protein, dFMRP, is found in adults’ brains, eyes, and mushroom bodies [21,49,50,51,52]. Some studies have also described dFMRP in *Drosophila* embryos and larvae [21]. The function of dFMRP in neuronal physiology, development, and structure is extensively studied in fruit fly larvae and adults. According to FlyBase data, dFMRP is involved in more than 50 biological processes with neuronal and non-neuronal functions in *Drosophila* [21]. The most important of these is its critical role in synaptic plasticity [21]. dFMRP plays an important role in aging, apoptosis, phagocytosis, and many other processes, too [21]. Figure 1 shows a schematic representation of the main localizations and the functions of dFMRP.

The range of genetic tools available when using *Drosophila* is greater than for any other multicellular organism, and there is a wide repertoire of available genetic manipulations [12]. Point mutations in *dfmr1* or the absence of all or most of the coding region of the *dfmr1* gene form the basis of the *Drosophila* FXS model [49,50,53,54,55]. Therefore, the first *dfmr1* mutant, i.e., the *Drosophila* FXS model, was generated by inaccurately excising a P element upstream of the *dfmr1* locus [49].

*dfmr1* mutants exhibit defective neuronal architecture and synaptic function. Furthermore, these mutants exhibit abnormally organized synapses in the peripheral and central nervous systems. The absence of dFMRP causes marked synaptic overgrowth at the neuromuscular junctions of *Drosophila* larvae [49,52,56,57]. The larval crawling pattern is also altered in *Drosophila* FXS models [58]. In another study, *dfmr1* larvae showed reduced motility during reorientation but normal motility during active crawling [59]. In addition, *dfmr1* mutants show altered behaviors such as irregular circadian rhythms, decreased male courtship activity, increased locomotion, learning and memory deficits, autism-like behaviors such as abnormal grooming, and social deficits [50,60,61,62,63]. Since the symptoms of FXS may be linked to the phenotypes of *dfmr1* mutants [21], the fruit fly model of FXS (Figure 2) is a robust model for studying FXS [22].

Here we have discussed the pathological phenotype observed in humans with FXS. Figure 3 shows the main behavioral phenotypes in humans with FXS and the overlapping loss-of-function phenotypes in *dfmr1* mutants.

### 3.1. Impaired Circadian Rhythms and Sleep Problems in FXS

Sleep problems are common in individuals with FXS. To illustrate, between 23% and 46% of individuals (predominantly younger males) with FXS have relatively mild sleep difficulties [43]. Another study found that the prevalence of sleep problems in individuals with FXS and ASD is about 80% higher than in the general population [64]. These problems may lead to circadian fluctuations and altered glucose homeostasis in individuals with FXS [65]. A consistent behavioral abnormality in the FXS model of *Drosophila* is altered circadian rhythm behavior (reviewed in [21]). The altered circadian rhythm in this animal model potentially mimics the sleep abnormalities observed in patients with FXS [50]. Circadian rhythms in *Drosophila* models manifest in locomotor activity, sleep/wake patterns, and various physiological and metabolic processes. Xu and colleagues (2012) described a role for *dfmr1* in miRNA pathways, pointing to the altered expression of selected miRNAs in the circadian abnormalities associated with the loss of *dfmr1* in flies [66]. Furthermore, Bushey and colleagues described that *dfmr1* regulates sleep needs in *Drosophila* FXS models [67].

A study by Dockendorff et al. (2002) showed that pupae from *Drosophila* larval control groups retained their ability to eclose. They were kept under 12 h of light/12 h of darkness conditions (for five days and then in complete darkness for six days), mostly locked into their circadian gap early in the morning with a duration of 23.5 h. In contrast, *Drosophila* FXS models under these conditions generally locked in later in the day and had a longer eclosion time with lower amplitude [54]. In another study, eclosion time in the *dfmr1* mutant remained dependent on the circadian rhythm, which was delayed by 6–8 h [50]. In contrast, Inoue et al. (2002) showed that the emergence of *dfmr1*^B55^ pupae was dependent on the circadian rhythm, with a similar phase and amplitude to wild type (*wt*) [53]. This inconsistency could be explained by the discovery made by Sekine et al. (2008) that the variations in eclosion time in FXS model flies are a consequence of the genetic background and not the deletion itself. When *dfmr1*^B55^ mutants were backcrossed to Canton S (CS) and yellow white (YW) flies, mutants with the same deletion from different genetic backgrounds were produced. These mutants did not exhibit eclosion abnormalities, suggesting that the eclosion deficit was due to the genetic background [68]. In addition, another study showed that several different types of *Drosophila* FXS models often do not survive because they fail to eclose. This phenomenon was particularly dominant in the *dfxr*^Δ50^ and *dfxr*^Δ113^ alleles, where 99% of flies failed to hatch [50].

To investigate circadian rhythms in more detail, the investigators conducted similar studies to see if the FXS flies exhibited circadian defects in locomotor activity. Interestingly, the locomotor activity of the *Drosophila* FXS models under 12 h light/12 h dark was similar to that of the control groups. However, once the flies in the control groups became accustomed to the regular alternation of day and night, they could maintain their rhythmic locomotor behavior for a certain period, even when walking freely in complete darkness. In contrast, the FXS models lacked this ability [4,50,53,69]. Notably, 20–30% of FXS models remained rhythmic despite constant darkness [68]. When *dfmr1*^B55^ mutants were backcrossed to CS and YW *Drosophila* strains for seven generations, the mutants maintained arrhythmic behavior in both genetic backgrounds, demonstrating that in this case, the genetic background does not affect circadian defects [68]. Nevertheless, a recent study challenged previous findings on circadian rhythms, suggesting that they may be influenced by excessive grooming in FXS *Drosophila* [70].

*Drosophila* FXS models have prolonged sleep compared to *wt* primarily due to an increased number of sleep episodes. On average, females slept 3 h longer and males 4 h longer than flies from control groups. After the end of the dark period, they woke up later than the control groups and slept longer during the day. In addition, their sleep was deeper, and they were less likely to wake up briefly [67]. Even their naps during the day were unusually deep and resembled night sleep [71]. In addition, null mutants showed defects in recovery from sleep deprivation by having shorter sleep episodes and a lower amount of recovered sleep. In contrast, overexpression of *dfmr1* results in shorter sleep [67].

Nowadays, there are a variety of software and systems for tracking locomotor activity and circadian rhythms in small model organisms such as *Drosophila*. The circadian rhythms of flies may be measured by recording their locomotor activity over time, and they are normally entrained by light intensity. Such advanced systems can be used to measure the activity of experimental models and also control (and change) the lighting regime. Figure 4 shows an example of tracking *Drosophila* in one of these types of software.

### 3.2. Hyperactivity and Attention Deficit/Hyperactivity Disorder in FXS

Attention deficit/hyperactivity disorder (ADHD) is one of the most common behavioral problems in individuals with FXS [39,72]. Thus, hyperactivity of locomotion has been used to assess drug efficacy in preclinical studies with animal models of FXS. In addition, hyperactive locomotion/climbing in a *Drosophila* model can mimic hyperactivity or ADHD in individuals [73,74,75].

The measurement of overall activity in the *Drosophila* FXS model has been inconsistent across studies. According to Dockendorff et al. (2002), total activity in *Drosophila* FXS models was not significantly different from control groups [54], but in another study, Morales et al. (2002) found that it decreased [50]. Another study showed lower locomotor activity in the *dfmr1*^B55^ mutant than in the *wt*. *dfmr*^B55^ by covering the space less well overall and making more stops (mostly at specific points) than the *wt*. In the same study, *dfmr1*^3^ covered the space more evenly but had more stops than the *wt* [62]. In addition, *Drosophila* FXS models had defects in flight ability, which were measured in the flight experiment [49]. An interesting finding linking *Drosophila* FXS models to FXS in humans and its model in *Fmr1* KO mice was relatively high bursts of activity. This behavior in *Drosophila* may correspond to the hyperactivity in the *Fmr1* KO mouse and humans with FXS [54].

One study examined changes in climbing ability in *Drosophila* FXS models with aging. Flies with a deletion in *dfmr1* were compared with two strains commonly used for laboratory purposes and one strain specific for its longevity. The climbing performance of the *Drosophila* FXS models decreased dramatically with age. Interestingly, the fastest 5-day-old *Drosophila* FXS models showed the same performance as other genotypes studied, but their climbing ability dramatically reduced at 25 days of age after eclosion. In contrast, the other genotypes gradually declined with age. *Drosophila* FXS models were found to be poor climbers at all ages in population studies and had the highest failure rate in completing the task among the flies studied [75].

### 3.3. Other Autistic-Like Behaviors in FXS

It is known that 50–60% of boys and 20% of girls with FXS are diagnosed with ASD [72,76]. Here we have discussed additional behavioral phenotypes that occur in the *Drosophila* FXS model that overlap with autistic phenotypes in individuals with FXS.

*Social interaction in the Drosophila FXS model*. Impaired social interactions are a characteristic feature of ASD [77]. The study’s results examining social interaction between two female *Drosophila* FXS models and their interaction with the *wt* model indicated normal receptive but altered expressive social behavior in the *Drosophila* FXS models. One possible explanation is that the models do not exhibit appropriate motor behavior or chemical signals necessary for social interaction [62]. Interestingly, *dfmr1^3^* did not spend much time at the boundary, regardless of whether the other chamber contained a mutant fly of the same type or the wt fly. Measuring the distance between the flies and the possibility that two flies were less than 5 mm apart confirmed these results [62].

*Grooming in the FXS model of Drosophila*. Excessive grooming in *dfmr1* mutant flies appears to reflect the hyperactive and ASD-like features of FXS observed in mice and humans [78].

According to the study data, the frequency and duration of grooming were significantly higher in 1-day-old mutants than in *wt*. The grooming index, i.e., the time spent grooming during a given time interval, was also significantly increased. The grooming pattern was altered such that the mutants groomed the posterior parts excessively, while the grooming of the anterior parts was similar to *wt* [70]. In another study, grooming increased progressively with age [78]. Moreover, excessive grooming was found to be persistent and structured. Namely, in the mutants, the switching between grooming pairs was excessively repeated during a run. Moreover, these FXS flies tended to start another grooming session with the same body part they had finished in the last session. Perseverative grooming corresponds to repetitive and stereotypical behaviors characteristic of ASD [70].

*Courtship behavior in the FXS model of Drosophila*. Inappropriate courtship behavior is described in individuals with ASD [79]. Naive courtship behavior is also altered in *Drosophila* FXS models [54,80,81,82,83]. Indeed, courtship in *Drosophila* consists of several phases leading to copulation. The male orients himself toward the female and follows her, then taps her with his legs, vibrates with his wing, licks her genital region, and finally attempts copulation [84]. Naive courtship behavior is estimated by the courtship index (CI), the percentage of time the male spends courting the female. In the *Drosophila* FXS models, the CI index was reduced, and they could not sustain courtship sufficiently to achieve more advanced phases of courtship, such as wing vibration, genital licking, and copulation attempts [54]. In the normal *Drosophila* population, other males sometimes court immature males. Still, FXS males courted immature males for less time than the control groups and persisted for less time in advanced phases [54]. The fact that immature male pheromones differ from female pheromones led to the conclusion that the courtship defect is not the result of specific sensory defects that may be present in *Drosophila* FXS models [54]. Furthermore, in one study, naïve courtship was impaired in both young and aged *Drosophila* FXS models, but the difference from the control groups was less marked in aged *Drosophila*. The explanation for this finding is naïve courtship also decreases with age in *wt* flies [80].

### 3.4. ID in FXS and Learning and Memory Impairment in the FXS Model of Drosophila

It is known that FXS is the leading cause of the inherited form of ID, and the deficit/absence of FMRP has been described as the core cause of ID in FXS. Although the relationship between *FMR1* expansion, gene methylation, and FMRP deficit is well known, the relationship between FMRP and ID needs more studies [85,86]. Only some assays exist to study learning and memory in *Drosophila* FXS models.

Almost 20 years ago, McBride and colleagues (2005) were the first to study learning and memory in *Drosophila dfmr1* mutants. They used the courtship paradigm to study associative memory. The complicated procedure of the experiment involved one hour of training a virgin male with an unresponsive female. Continuous rejection should teach memory-intact males to stop courting. Time spent courting was reduced in *dfmr1* mutants and control groups, implying that FXS flies can learn during training. This form of memory is a mixture of associative and non-associative memory [82]. Although learning is preserved in young FXS flies, FXS models lose this function at 20 days of age when tested in the same assay [80]. To examine only associative memory, these researchers placed flies that had learned to be rejected in the chamber with receptive females. Tests were performed immediately after training. The results showed that the males in the *Drosophila* FXS models forgot what they had learned and attempted to copulate as often as the control groups. This study was the first to identify defects in immediate recall memory, memory lasting 0–2 min after training, and short-term memory that lasts up to one hour after training in FXS *Drosophila* [82]. Long-term memory can also be examined using tests based on the courtship paradigm. Using this test, Banerjee et al. (2010) demonstrated that long-term training memory is impaired in FXS models of *Drosophila* [81].

Another popular method for assessing memory in *Drosophila* is based on classical conditioning. In this test, flies were exposed to two odors. The first odor was followed by an electric shock, while the second was not. The trained flies were then placed in a T-maze to choose between the previously exposed odors. Learning and memory were assessed in the next step by measuring the percentage of flies in the chamber with an odor that was not followed by the electric shock. Learning was assessed immediately after training, and memory after a while, depending on the type of memory [69]. Caffe et al. (2012) found that learning ability decreased significantly in FXS models of *Drosophila* compared to *wt* [87]. This test could estimate long-term memory when performed repeatedly. When the training was performed ten times without interruption, a decremental cycloheximide-insensitive memory (ARM) was formed. The same procedure, with a 15-min break between training sessions, would also develop ARM, but non-decremental cycloheximide-sensitive long-term memory (LTM) was also developed. The latter form of memory was dependent on protein synthesis. When *Drosophila* FXS models were placed in the T-maze one day after repeated training, they showed defects in LTM but not ARM. Finally, silencing of *dfmr1* only in mushroom bodies showed the same results [61].

## 4. *D. melanogaster* as a Model Organism to Study Fragile X-Associated Tremor/Ataxia Syndrome (FXTAS)

As described above, the normal range of CGG repeats in the human *FMR1* gene is between 5 and 44. The premutation range (PM) of the *FMR1* gene is characterized by 55–200 CGG repeats, and those carriers typically do not present with FXS symptoms. However, they are associated with three other disorders: Fragile X-associated primary ovarian insufficiency (FXPOI) [88], Fragile X-associated tremor/ataxia syndrome (FXTAS) [89,90], and Fragile X-associated neuropsychiatric disorders (FXAND) [91].

FXTAS is a progressive neurodegenerative disorder that occurs in approximately 40% of males and 16% of female carriers. Individuals diagnosed with FXTAS present with a progressive intention tremor, difficulty with ambulation, ataxia, deficits in executive function, and brain atrophy associated with elevated *FMR1* mRNA levels [90,92]. The prevalence of FXTAS increases with age. A study of PM men showed that 17% were affected at age 50, 38% at age 60, 47% at age 70, and 75% at age 80 [93]. In contrast to FM of *FMR1*, which results in transcriptional silencing of *FMR1* mRNA and a concomitant loss of FMRP, in FXTAS, there are normal FMRP levels or a modest reduction in the high PM repeat range. However, in PM carriers, there is a dramatic increase in *FMR1* mRNA levels, leading to mRNA toxicity and the pathogenesis of FXTAS [72,94]. Elevated mRNA levels lead to increased Ca^2+^ levels in the neuron and subsequent mitochondrial dysfunction. Proteins and RNAs are sequestered in inclusion bodies. The formation of R-loops leads to DNA damage. RAN translation leads to the production of toxic polyglycine-containing (FMRpolyG) proteins [72,94].

Jin and colleagues first described the *Drosophila* model of FXTAS in 2003. The PM range in *dfmr1* alone is sufficient to cause neurodegeneration [95]. Therefore, FXTAS has been modeled in *Drosophila* by overexpressing 90 CGG repeats fused with a green fluorescent protein (GFP), resulting in neuron-specific degeneration and inclusion formation [95,96]. Notably, FMRpolyG is toxic and directly influences the toxicity of CGG repeat constructs in *Drosophila* [97]. In addition, Jin and colleagues described that the CGG-induced neurodegenerative phenotype in the *Drosophila* FXTAS model could be rescued by overexpression of purα [98]. Other studies identified some tropomyosin and RNA-binding proteins as genetic modifiers of neurodegeneration in the *Drosophila* FXTAS model [98,99].

Flies with modest PM in *dfmr1* (rCGG_90_ repeats) exclusively in neurons do not reach adulthood. Lethality occurs mainly during embryonic development prior to larval formation [95]. This model shows deficits in locomotion and retinal degeneration [95].

## 5. Conclusions and Future Perspectives

Currently, there is no cure for FXS in patients, but the studies of medication use in *Drosophila* have been helpful in initiating medication core modifier clinical trials that work well in the fly and transitioning such treatments to humans with FXS. Examples of this include the minocycline studies in flies that were translated to humans, demonstrating behavioral benefits [100]. Another example is the benefit of metformin in flies [101], which led to clinical trials in humans with FXS [102,103,104,105]. This review has emphasized the simplicity and cost-effectiveness of preclinical trials in flies to encourage young researchers to further the studies of new targeted treatments in flies that can be translated to patients with FXS. Since the lack of FMRP has profound effects on many systems paralleled in flies and humans, this animal model will be utilized many times in the future to guide new treatments for FXS and possibly FXTAS.

## Figures and Tables

**Figure 1 genes-14-00087-f001:**
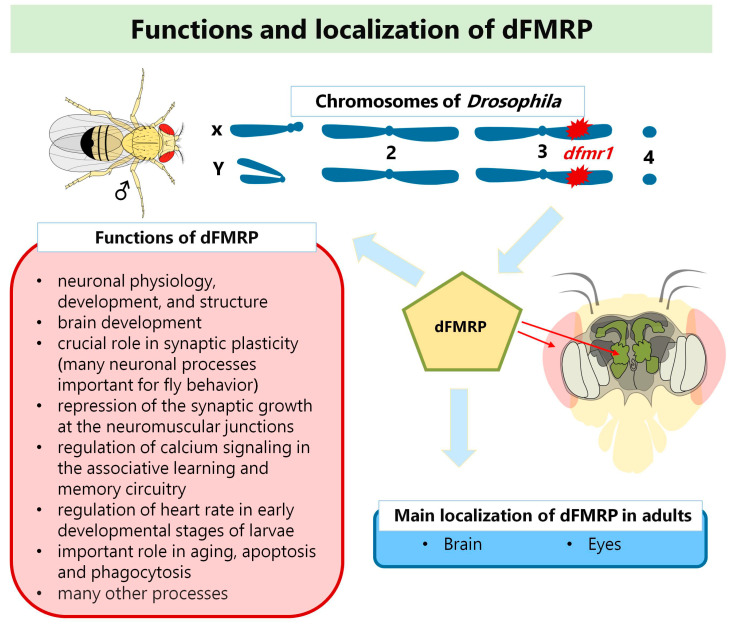
Functions and localization of dFMRP in *D. melanogaster* (adapted from reference [21]) Aberrations: *dfmr1*-Fragile X Messenger Ribonucleoprotein 1 gene in *Drosophila*; dFMRP-*dfmr1* protein in *D. melanogaster.*

**Figure 2 genes-14-00087-f002:**
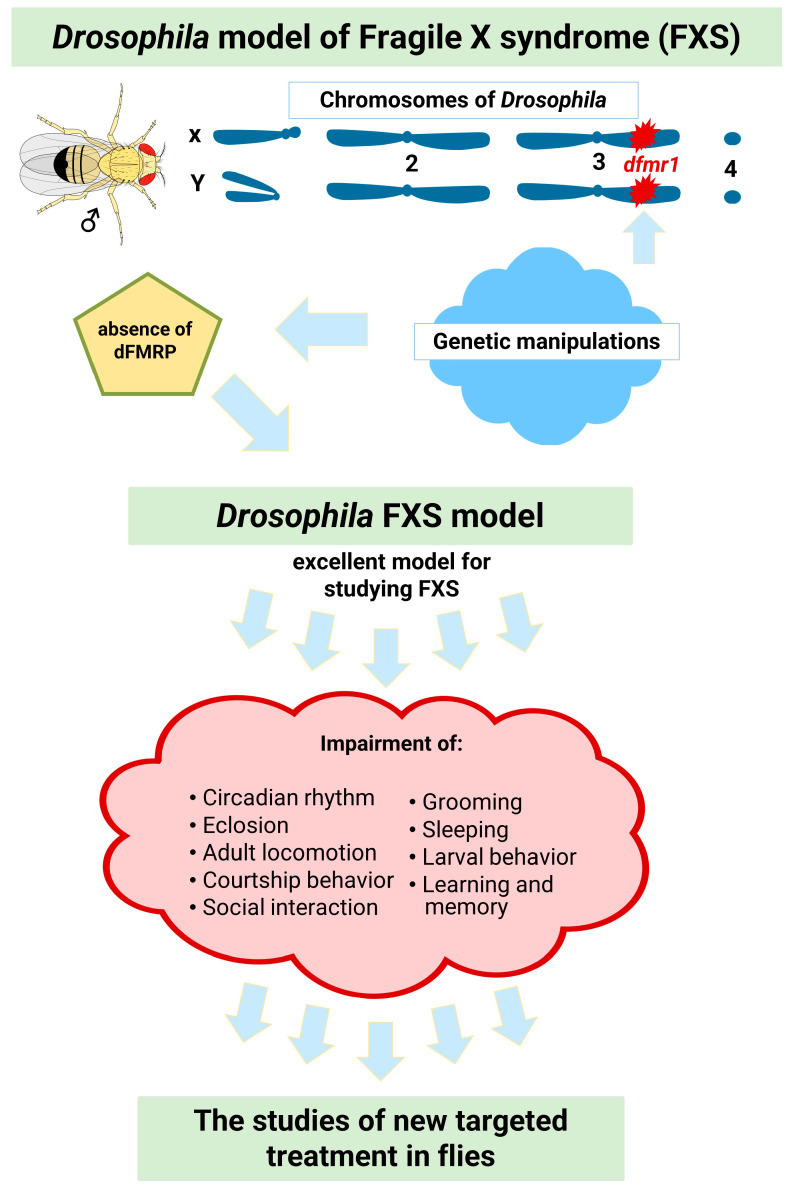
*Drosophila* model of fragile X syndrome (adapted from reference [21]). Aberrations: *dfmr1*-Fragile X Messenger Ribonucleoprotein 1 gene in *Drosophila;* dFMRP-d*fmr1* protein in *Drosophila*).

**Figure 3 genes-14-00087-f003:**
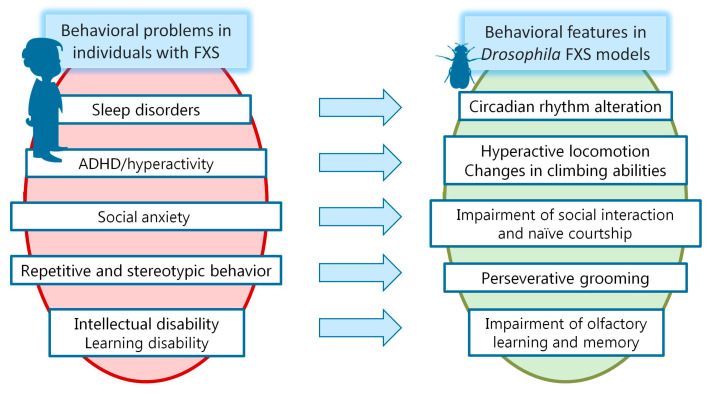
Major phenotypes in humans with FXS and overlapping loss-of-function phenotypes in *dfmr1* mutants.

**Figure 4 genes-14-00087-f004:**
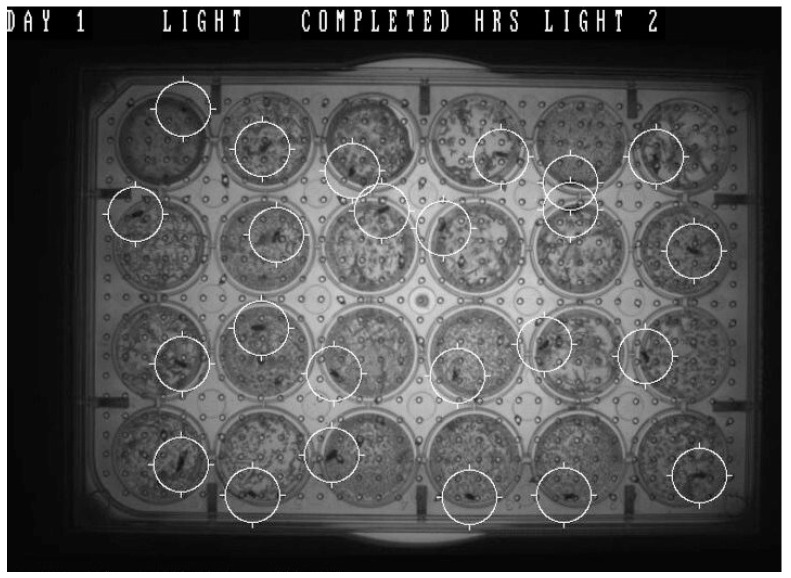
An example of the study of circadian rhythms in *Drosophila* using the software. (The image is from Protic’s lab: www.polyfrax.com, accessed on 1. December 2022.). A multi-well plate was prepared. Depending on experimental needs, the plates can be 24-, 48-, or 96- well. The flies were placed individually in each well, and an air-permeable cover was put over the top of the well plate. The flies were then individually housed in a humid environment with some available nutrition. The well plate was loaded with flies and inserted into the system’s chamber. The software was set up to control the environment (temperature and lighting) and the measure of the distance traveled by each fly. The white cycles show the flies as software targets in 24 wells. The activity (measured as distance traveled) is measured and recorded as data during the experiment.

## Data Availability

Not applicable.

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
