# Peer review of "Drosophila melanogaster as a Model to Study Fragile X-Associated Disorders"

_genes, 2022, doi:10.3390/genes14010087_

Round 1

Reviewer 1 Report

In this review article, "Drosophila melanogaster as a model to study Fragile X syndrome," the authors attempt to detail (i) how Drosophila can be used as a model organism to model human diseases and (ii) a summary of pathologically relevant behavioral phenotypes in Drosophila model of fragile X. Fragile X Syndrome (FXS), a hereditary neurodevelopmental disorder, most frequently caused by CGG trinucleotide repeat expansion in the 5' of the Fragile-X loci in Xq27.3, which encodes FMR1 protein (FMRP). Animal models of FXS, such as mice, rats, zebrafish, and Drosophila, provided more precise insights into human disease pathogenesis. 

While this review is timely, the article is limited to a comprehensive summary of understanding the fragile X syndrome using the Drosophila model. The authors mainly focus on the behavioral defects reported in Drosophila FX mutants. Considering the previous publications – Drozd et al., (2018); McBride et al., (2012); Specchia et al., (2019) that review Drosophila studies investigating FMRP functions at genetic, cellular, molecular, electrophysiological, and behavior levels, and also research on pharmacological treatments in the fly model, it remains unsure whether the current review article makes any significant contribution. Scientific content, including the flow, avoiding the jargon, citing appropriate literature, etc., the manuscript needs to be improved. 

Authors might consider the below discussed points to improve the manuscript.

1. The title of the article, "Drosophila melanogaster as a model to study Fragile X syndrome," aims to discuss the recent studies in Drosophila FX models. However, the author provides a detailed overview of Drosophila in biomedical research, emphasizing the historical milestone in fruit fly research. Rather than that, the author should focus on the conserved FMRP functions, molecular mechanisms, and how the Drosophila FX model contributed to our understating of the FXS. FMRP function in neuronal physiology, development, and structure are extensively investigated in fruit fly larvae and adults that need to be discussed.

2. The article could be easier to read (readability of the current version is harder for researchers from fruit fly or fragile-X backgrounds), if the manuscript includes appropriate background information on experimental phenotypes discussed in the review. The authors' oversight of the specific details is evident throughout the manuscript. To quote an example (lines 246 – 248): "One study investigated changes in climbing ability in Drosophila FX models with aging. Flies with a deletion in dfmr1 were compared with two commonly used strains for laboratory purposes and one strain specific for its longevity." What are the strains the authors are referring to? Additionally, it would be helpful if the authors discussed fly behaviors in light of the pathological phenotype reported in humans with FXS.

3. Manuscript needs to be edited for English writing and typographical errors. A few examples are below: 

a. Lines 205 and 206, enclosed should be replaced with eclosed.

b. Line 208, faze-delayed. Do the authors mean Phase-delayed?

c. Line 256 to 259: "Naive courtship is altered in Drosophila FX mutants [78, 89-92]. Namely, courtship in Drosophilaconsists of several phases that led to copulation. Male is oriented towards the female and follows her, then taps her with his legs, vibrates with his wing, licks her genital region and finally tries to copulate [93]."

4. In the section on genetic manipulation of studying FXS using Drosophila, the authors explain the p-element mutagenesis and CRISPR technologies per se. It would be relevant if the authors explained how these technologies are employed in generate FX mutants. 

5. Two other disorders: Fragile X Associated Tremor/Ataxia Syndrome (FXTAS) and Fragile X Primary Ovarian Insufficiency (FXPOI), are reported in individuals with 55-200 CGG repetitions who don't show FXS symptoms. In addition to FXS, FXTAS has also been modeled in Drosophila by overexpressing 90 rCGG repeats alone fused to GFP, (Jin et al., 2003; Qurashi et al., 2012). The authors need to discuss FXTAS in this review article. 

6. A table summarizing the observed phenotypes of dFMR1 mutants and the schematic of the dFMR1 signaling network would be helpful.  

Reviewer 2 Report

line 33 "protein (FMRP), is located on chromosome Xq27.3. There are between 5 and 44 CGG " may be modified as "protein (FMRP) is located on chromosome Xq27.3 between 5 and 44 CGG........."

Lines 59-109 may be condensed as a small paragraph and included in the introduction itself

Lines 119 and 149 are also generalised in nature not closely related to the theme of the review and may be condensed and those parts not related to the actual concept mat be removed

line 153 cited only one reference and indicated that there are wide repositories, hence more cited reference may be included

line 156-157: need modification not conveying correct meaning

line 274 "The results indicate normal receptive, but altered expressive" may be modified as "The results indicated that the  normal receptive, but altered expressive...."

The conclusion part is not comprehensive.  

Round 2

Reviewer 1 Report

In the revised version, the authors have improved the scientific content of the review. However, there is huge scope to improve the clarity and comprehensive detailing of Drosophila fmr1 literature. 

1. Manuscript needs to be edited for spelling and typographical errors. 

2. Figure 1 is not very helpful. Also, some information - fmr1 mainly expressed in PNs or MB of the brain is misleading. 

3. Still determining the relevance of Figure 4 for this review! Instead, the authors might consider the FMRP molecular signaling and interacting network schematic.

4. Gene and scientific names should be italicized.

Reviewer 2 Report

Dear Sir

The article has been modified as per the suggestions and may be recommended for publication

kr

akt

Author Response

Dear Reviewer,

Thank you very musch.

Best regards,

Dragana Protic, MD, PhD